# The Impact of Heavy Perceived Nurse Workloads on Patient and Nurse Outcomes

**Maura MacPhee \*, V. Susan Dahinten, and Farinaz Havaei**

The University of British Columbia School of Nursing, Vancouver, BC V6T 2B5, Canada;
Susan.Dahinten@nursing.ubc.ca (V.S.D.); farinazhavaei@gmail.com (F.H.)
\* Correspondence: Maura.MacPhee@nursing.ubc.ca; Tel.: 001-778-926-4068

**Abstract:** This study investigated the relationships between seven workload factors and patient and nurse outcomes. (1) Background: Health systems researchers are beginning to address nurses' workload demands at different unit, job and task levels; and the types of administrative interventions needed for specific workload demands. (2) Methods: This was a cross-sectional correlational study of 472 acute care nurses from British Columbia, Canada. The workload factors included nurse reports of unit-level RN staffing levels and patient acuity and patient dependency; job-level nurse perceptions of heavy workloads, nursing tasks left undone and compromised standards; and task-level interruptions to work flow. Patient outcomes were nurse-reported frequencies of medication errors, patient falls and urinary tract infections; and nurse outcomes were emotional exhaustion and job satisfaction. (3) Results: Job-level perceptions of heavy workloads and task-level interruptions had significant direct effects on patient and nurse outcomes. Tasks left undone mediated the relationships between heavy workloads and nurse and patient outcomes; and between interruptions and nurse and patient outcomes. Compromised professional nursing standards mediated the relationships between heavy workloads and nurse outcomes; and between interruptions and nurse outcomes. (4) Conclusion: Administrators should work collaboratively with nurses to identify work environment strategies that ameliorate workload demands at different levels.

**Keywords:** nursing workload; patient adverse events; nurse outcomes; nursing tasks left undone; interruptions; nurse staffing; compromised professional nursing standards

---

## 1. Introduction

The global RN4CAST consortium (http://www.rn4cast.eu/en/consortium.php) was established to support the accuracy of forecasting models and generate new approaches to more effective management of nursing resources across countries. The global RN4CAST project with over 11,000 patients and 33,000 nurses demonstrated that, regardless of country, when nurses have heavy workloads, they leave essential tasks undone, and there are negative nurse and patient outcomes [1,2]. Understanding workload and its impact, particularly from nurses' perspectives, is an urgent undertaking, given global nurse shortages and the associations between workload and nurse retention [2,3].

### 1.1. Workload Considerations

There is no common definition for nurses' workload. Workload is often associated with the volume of nurses' work, and there have been many attempts to quantify nurses' work in relation to health human resource management [4]. We were interested in identifying key predictors that can be used to identify worrisome trends and avert serious outcomes, such as patient mortality and

morbidity. The notion of leading and lagging indicators was recently discussed in a paper by Ball et al. that urged employers and regulators to focus on leading nurse indicators that have the potential to proactively address quality and safety deficiencies [5]. Our study goal, therefore, was to focus on nurse-perceived workload factors that are assessable and actionable. In our study, we included workload factors from a variety of validated, publicly available assessment tools, particularly those used in the global RN4CAST studies [6]. We were also influenced by the human factors framework of Holden et al. [7].

Human factors frameworks are becoming popular ways to examine nurses' workloads at three different levels—unit-level, job-level, and task-level [7]. Unit-level workload includes staffing level and skill mix considerations; job-level workload is based on nurses' perceptions of the "general amount of work to be done in the day" [7], (p. 15); and task-level workload considers the nurse resources to do a task, such as mental concentration associated with medication administration. Each workload level is associated with different cognitive demands and nurse and patient outcomes. For example, medication errors are best predicted by task-level demands [7]. A comprehensive appreciation of nurses' workloads, therefore, requires assessment of nurse workload demands at all three levels.

One conceptualization of nursing workload at the unit-level is patient care intensity [8]. Assessment of patient care needs underpins nursing workload measurement, and there are a variety of patient assessment or classification systems within the literature. Most systems focus on acuity or severity of illness; or dependency, the need for support with activities of daily living [8,9]. Nurse assessment has been used to determine patient acuity and patient dependency needs [10,11].

At the unit-level, nursing workload is also commonly measured by staffing levels or patient-nurse ratios. A systematic review of 102 studies demonstrated that increased registered nursing (RN) staffing levels were associated with decreased rates of mortality in medical-surgical settings [12]. This association was supported by a later review with 15 new primary studies [13]. Despite compelling evidence that there is a link between RN staffing levels and patient outcomes, such as mortality, the pathway(s) by which staffing levels influence outcomes is not well understood. Griffiths et al. also pointed out that after 20 years of research on nurse staffing, "the role of mechanisms in the causal path [through which nurse staffing can influence outcomes] has rarely been directly demonstrated through studies…" [14] (p. 24). Some moderating/mediating mechanisms that have been explored to date include missed care [5] and care left undone [15].

Ball et al. postulated that when care is not done or "missed", the quality and safety of patient care may be compromised [5]. Based on the RN4CAST protocol, Ball et al. surveyed National Health Service England nurses about job-level care left undone on their most recent shift worked for 13 essential, nursing care activities. On average, nurses reported leaving four care items undone on their most recent shift. A frequent missed care item was patient surveillance, or the capacity to monitor patients for status changes [5]. Ball et al. found significant associations between nurses' reports of missed care, RN staffing levels, and perceptions of patient care quality [5,16]. These authors surmised that missed care may be a job-level "leading indicator" for identifying quality of care deficiencies before there are serious consequences, such as unnecessary loss of life [5].

Myny et al. identified factors affecting nursing workload by conducting an integrative literature review, and then determining relevance and measurability of these factors through focus groups and a survey [17]. The factor with the highest workload "impact score" was "high number of work interruptions". Work interruptions at the task-level negatively influence cognitive or mental load, leading to emotional duress and error. Since a significant component of RNs' work is knowledge work, competencies associated with assessment, analysis, synthesis and coordination, are compromised by unanticipated interruptions [18]. In a Canadian study of RN interruptions on medical-surgical units, almost one-third of interruptions occurred during patient assessments and procedures, while another one-third occurred during patient documentation [19]. These authors concluded that 89% of observed interruptions had the potential to adversely impact patient safety.

*1.2. Workload Outcomes*

Commonly measured nurse outcomes include job dissatisfaction and burnout. Burnout has been linked to higher rates of absenteeism than the general population [20], and to increased nurse turnover [21] and decreased job satisfaction [22]. Leiter and Maslach and Kowalski et al. found that heavy perceived nurse workloads were associated with one component of burnout, emotional exhaustion [23,24]. Without adequate resources and supports to meet workload demands, nurses grow dissatisfied and emotionally exhausted; they burn out and leave--sometimes leaving the profession altogether [21]. Holden et al. found that nurse job satisfaction was positively associated with a unit-level workload measure, staffing adequacy; burnout was negatively associated with unit-level staffing adequacy, and positively associated with task-level external demands, such as interruptions [7].

Greater nursing workloads are associated with adverse patient outcomes [25]. Globally, researchers have used nurse-sensitive adverse patient outcomes to study the relationships between nurses' work environments, their workloads and patient outcomes [26]. Nurse reports of patient adverse events are often used as a proxy for administrative unit-level data (i.e., actual morbidity, mortality rates), because accurate unit-level data are difficult to obtain. Although nurse reports of patient adverse events are prone to recall bias, some research has established concordance between nurse reports and actual patient adverse events, such as falls with injuries [27]. For the RN4CAST studies, nurse ratings of unit-level quality of care included estimates of frequency of patient adverse events, such as medication errors, falls and hospital-acquired infections [6].

## 2. Objectives

The aim of this study was to understand the effect of unit, job and task-level workload factors on three adverse patient outcomes (medication errors, patient falls, and urinary tract infections) and two nurse outcomes (emotional exhaustion and job satisfaction). We considered seven workload factors: unit-level RN staffing levels, patient acuity, patient dependency, job-level nurse perceptions of heavy workload, tasks left undone, compromised professional nursing standards, and task-level interruptions. In addition, we tested the potential mediating effects of two variables: nursing tasks left undone and compromised professional nursing standards.

We asked the following research questions:

(1) What are the relationships between perceptions of heavy perceived nurse workload, interruptions to workflow, nursing tasks left undone, and compromised professional nursing standards and the frequency of (a) medication errors; (b) patient falls; and (c) urinary tract infections after accounting for RN staffing levels and patient acuity and patient dependency?

(2) What are the relationships between perceptions of heavy perceived nurse workload, interruptions to workflow, nursing tasks left undone, and compromised professional nursing standards and nurses' (a) emotional exhaustion; and (b) job satisfaction after accounting for individual characteristics, RN staffing levels, and patient acuity and patient dependency?

(3) Are the effects of perceptions of heavy perceived nurse workloads and interruptions to workflow on the three patient outcomes mediated by nursing tasks left undone and compromised professional nursing standards?

(4) Are the effects of perceptions of heavy perceived nurse workloads and interruptions to workflow on the two nurse outcomes mediated by nursing tasks left undone and compromised professional nursing standards?

## 3. Materials and Methods

The data for this cross-sectional correlational study were extracted from a web-based survey on nurses' perceptions of their working environment, quality of nursing care, patient outcomes, and nurse outcomes among a province-wide sample of Canadian nurses. Institutional Review Board ethics was obtained (approval number: H14-00789). A proportionate stratified random sample of RNs and licensed practical nurses (LPNs) was drawn from the provincial nurses' union database based

on geographic region (i.e., health authority) and employment status (full-time, part-time, and casual). In Canada, RN and LPN classifications are distinguished by differences in formal education and scopes of practice. Registered nurses receive more theoretical education and are prepared to care for complex, unstable patients, while LPNs are prepared to care for stable, predictable patients. The survey was content validated by union member focus groups. Unique, password-protected FluidSurvey email invitations were sent out by the nurses' union on behalf of the research team.

The study sample consisted of all direct care nurses working in medical, surgical or medical-surgical areas in the four largest health authorities. All direct care nurses in acute care settings in British Columbia (BC) are unionized; therefore, we had a complete sample frame. Our final sample (*N* = 472) consisted of 354 RNs and 118 LPNs with an estimated response rate of 22.4%. Precise response rates were difficult to determine due to the nature of the union's database (e.g., active versus inactive members). A similar issue is noted by Ball and colleagues [5].

## 3.1. Measures

*Adverse Patient Outcomes* were measured using RN4CAST questions that asked nurses to estimate the frequency of adverse events (i.e., medication errors, patient falls, and urinary tract infections) "involving you or your patients" on a scale ranging from 0 (*never*) to 6 (*everyday*) during the last year [6]. For this study, we recoded data as *occurred less than weekly* (0) versus *occurred weekly or more often* (1).

*Emotional Exhaustion* among nurses was measured with the 9-item subscale of the Maslach Burnout Inventory–Human Service Scale (MBI-HSS) [28]. The emotional exhaustion subscale asks participants to rate their work-related feelings of psychological depletion on a scale of 0 (*never*) to 6 (*daily*). For this study, the total scores (ranging from 0–54) were dichotomized with scores of 27 and higher indicating high emotional exhaustion or burnout per developer instructions [28].

*Nurses' Job Satisfaction* was measured as the sum of three variables that asked about satisfaction with current job, intent to leave current job during the next year (reverse coded), and recommending the hospital to colleagues as a good place to work. Each item was measured on a 4-point scale. Total scores ranged from 3–12 with higher scores indicating greater job satisfaction. These items were derived from the validated Canadian National Survey on the Work and Health of Nurses [29].

*RN Staffing Levels* were measured by computing a patient-to-RN ratio based on two questions that asked nurses to identify the total number of patients and total number of direct care nursing staff on the unit during their last shift. Patient-to-RN ratio was used rather than the patient-to-nurse (RN or LPN) ratio for consistency purposes, as many units did not utilize LPNs. This staffing level method is described in Sermeus et al. [6].

*Patient Acuity and Patient Dependency* were measured with one item each based on the American Association of Critical Care Nurses' Synergy Model™ [30]. Patient acuity was defined as the instability, complexity, and unpredictability of the patient: participants were asked to rate the average acuity of their patients during the prior month from 1 (*not at all acute*) to 4 (*very acute*). For this study, we dichotomized acuity levels as *not at all or somewhat acute* (0) versus *moderately or very acute* (1). Patient dependency was defined as a patient's ability to do their own activities of daily living, rated from 1 (*very independent*) to 4 (*very dependent*). These scores were dichotomized as *very or somewhat independent* (0) to *very or somewhat dependent* (1).

*Perceptions of Nurse Workload* were measured as the mean score of three items that asked about the frequency of arriving early/staying late, working through breaks to complete work, and perceptions of "too much work" during the past year, measured on a scale of 0 (*never*) to 6 (*every day*). The mean scores were dichotomized as *never to a few times a week* (0) versus *occurring every day* (1). These items were taken from the Canadian National Survey on the Work and Health of Nurses [29].

*Nursing Tasks Left Undone* was measured by asking nurses to identify, from a list of 14 activities, all the activities that were necessary but left undone during their most recent shift due to lack of time; for a possible range of scores from 0 to 13. Thirteen nursing tasks were identified by Ball et al., including administering medications on time, preparing patients and families for discharge, and adequate patient surveillance [5]. We added an "other" option to our survey tool.

*Compromised Professional Nursing Standards* was measured with a single item that asked nurses the frequency of compromised professional nursing standards over the past year due to workload, measured on a scale of 0 (*never*) to 6 (*everyday*). Scores were dichotomized as *never to a few times a week* (0) versus *occurring everyday* (1). This item was added to reflect nurses' "meaning of work" [31]. Our researcher-developed question was content-validated with nursing focus groups.

*Interruptions* to workflow were measured as the mean score of three items that asked about the frequency of interruptions over the past month during patient treatments, during documentation, and when receiving patients at shift change, measured on a scale of 0 (*never*) to 6 (*everyday*). The mean scores were dichotomized as *never to a few times a week* (0) versus *occurring every day or almost every day* (1). These items were based on a focused literature review and content validation with nurse focus groups.

*Factor Structure* of each of the four measures that involved a mean score (perceptions of workload, interruptions to workflow, emotional exhaustion, and job satisfaction) were examined using exploratory factor analyses with principal components analysis; the results indicated a unidimensional factor structure and satisfactory internal consistency for all measures. Cronbach's alphas ranged from .67 for job satisfaction (with only 3 items) to .93 for emotional exhaustion (with 9 items). The percentage of variance explained by the single factor ranged from 62% for job satisfaction to 75% for interruptions to workflow.

*3.2. Data Analysis*

Data were analyzed using hierarchical logistic regression and hierarchical ordinary least squares regression according to the nature of the outcome variable, using the Statistical Package for Social Sciences for Windows 23.0 (SPSS Inc., Chicago, IL, USA). Mediation effects were tested using the Sobel Test [32], with adjustments made to the coefficients [33] for the inclusion of dichotomous mediator and outcome variables.

## 4. Results

Table 1 presents the demographic characteristics of the total sample of 118 LPNs and 354 RNs. Among the sample, 56% had a nursing degree. The average age of the predominantly female sample was 38.4 years among the RNs and 43.6 years among the LPNs ($t$ = 4.23, $p$ < 0.001). There were no significant differences between RNs and LPNs in employment status (i.e., full-time or less). Nor were there any statistically significant differences in the RN vs. LPN group's scores for perceptions of heavy workload, interruptions to workflow, or compromised professional nursing standards. More of the RN group (81%) assessed their patients' acuity as moderately or very acute compared with the LPN group (63%, $\chi^2$ = 15.2, $p$ < 0.001). Table 2 presents the descriptive characteristics for key variables in the study, and Table 3 presents the inter-correlations.

**Table 1.** Descriptive statistics of sample ($N$ = 472).

| Characteristic | M (SD) | f (%) |
|---|---|---|
| **Age** | 39.7 (11.8) | - |
| **Gender** | | |
| Male | - | 19 (4.1%) |
| Female | - | 449 (95.9%) |
| **Professional Designation** | | |
| Registered Nurse (RN) | - | 354 (75.0%) |
| Licensed Practical Nurse (LPN) | - | 118 (25.0%) |
| **Nursing Education** | | |
| Diploma or Certificate | - | 206 (43.6%) |
| Baccalaureate or Masters | - | 266 (56.4%) |
| **Employment Status** | | |
| Full-time | - | 276 (58.5%) |
| Part-time or Casual | - | 196 (41.5%) |

**Table 2.** Descriptive statistics for key predictors and outcome variables (*N* = 472).

| Predictors | M (SD) | f (%) |
|---|---|---|
| **Patient Acuity** | | |
| Not at All or Somewhat Acute | - | 110 (23.5%) |
| Moderately or Very Acute | - | 358 (76.5%) |
| **Patient Dependency** | | |
| Very or Somewhat Independent | - | 68 (14.6%) |
| Very or Somewhat Dependent | - | 399 (85.4%) |
| **Heavy Workload** | | |
| Never to a Few Times a Week | - | 351 (74.4%) |
| Everyday | - | 121 (25.6%) |
| **Interruptions** | | |
| Less than Almost Everyday | - | 299 (63.8%) |
| Every Day or Almost Everyday | - | 170 (36.2%) |
| **Compromised Standards** | | |
| Never to a Few Times a Week | - | 386 (81.8%) |
| Everyday | - | 86 (18.2%) |
| **Patient–RN Ratio** | 6.7 (3.2) | - |
| **Tasks Left Undone** | 4.8 (3.1) | - |
| **Outcomes** | | |
| **Medication Errors** | | |
| Less than Weekly | - | 405 (86.2%) |
| Weekly or More Often | - | 65 (13.8%) |
| **Patient Falls** | | |
| Less than Weekly | - | 415 (88.3%) |
| Weekly or More Often | - | 55 (11.7%) |
| **Urinary Tract Infections** | | |
| Less than Weekly | - | 406 (86.4%) |
| Weekly or More Often | - | 64 (13.6%) |
| **Emotional Exhaustion** | | |
| No (0–26) | - | 209 (44.6%) |
| Yes (27–54) | - | 260 (55.4%) |
| **Job Satisfaction** | 7.7 (2.2) | |

**Table 3.** Correlations between key study variables.

| Study Variables | 1 | 2 | 3 | 4 | 5 | 6 | 7 | 8 | 9 | 10 | 11 | 12 | 13 | 14 |
|---|---|---|---|---|---|---|---|---|---|---|---|---|---|---|
| 1. Age | — | | | | | | | | | | | | | |
| 2. Professional Designation [1] | −0.19 *** | — | | | | | | | | | | | | |
| 3. Employment Status [2] | −0.16 ** | −0.04 | — | | | | | | | | | | | |
| 4 Patient–RN ratio | 0.04 | −0.18 *** | 0.00 | — | | | | | | | | | | |
| 5. Patient Acuity [3] | −0.03 | 0.18 *** | −0.08 | −0.13 ** | — | | | | | | | | | |
| 6. Patient Dependency [4] | −0.13 ** | 0.04 | −0.02 | 0.02 | 0.06 | — | | | | | | | | |
| 7. Heavy Workload [5] | 0.08 | 0.03 | −0.04 | 0.02 | 0.13 ** | 0.07 | — | | | | | | | |
| 8. Interruptions [6] | 0.13 ** | −0.02 | −0.14 ** | 0.06 | 0.22 *** | 0.03 | 0.30 *** | — | | | | | | |
| 9. Tasks Left Undone | −0.01 | −0.11 * | 0.01 | 0.15 ** | 0.09 | 0.16 ** | 0.36 *** | 0.29 *** | | | | | | |
| 10. Compromised Standards [7] | 0.10 * | −0.06 | −0.06 | 0.07 | 0.10 * | 0.04 | 0.34 *** | 0.32 *** | 0.36 *** | — | | | | |
| 11. Medication Error [8] | 0.08 | −0.07 | −0.04 | 0.11 * | 0.17 *** | 0.04 | 0.22 *** | 0.22 *** | 0.30 *** | 0.22 *** | — | | | |
| 12. Patient Falls [9] | 0.13 ** | −0.05 | −0.04 | 0.20 *** | 0.14 ** | 0.06 | 0.32 *** | 0.28 *** | 0.35 *** | 0.24 *** | 0.41 *** | — | | |
| 13. Urinary Tract Infections [10] | 0.11 * | −0.06 | 0.01 | 0.15 ** | 0.13 ** | 0.09 | 0.26 *** | 0.25 *** | 0.25 *** | 0.13 *** | 0.27 *** | 0.69 *** | — | |
| 14. Emotional Exhaustion [11] | 0.04 | −0.07 | −0.10 * | 0.11 * | 0.17 *** | 0.03 | 0.34 *** | 0.32 *** | 0.38 *** | 0.33 *** | 0.23 *** | 0.23 *** | 0.21 *** | — |
| 15. Job Satisfaction | −0.04 | −0.05 | −0.02 | −0.12 * | −0.11 * | 0.06 | −0.35 *** | −0.28 *** | −0.38 *** | −0.42 *** | −0.29 *** | −0.30 *** | −0.22 *** | −0.51 *** |

Note: [1] 0 = LPN, 1 = RN; [2] 0 = full-time, 1 = part-time or casual; [3] 0 = not at all or somewhat acute, 1 = moderately or very acute; [4] 0 = very or somewhat independent, 1 = somewhat or very dependent; [5] 0 = never to a few times a week, 1 = more than a few times a week; [6] 0 = less than every day, 1 = every day or almost every day; [7] 0 = less than every day, 1 = every day; [8] 0 = less than weekly, 1 = weekly or more often; [9] 0 = less than weekly, 1 = weekly or more often; [10] 0 = less than weekly, 1 = weekly or more often; [11] 0 = no burnout, 1 = burnout. $*p < 0.05$, $** p < 0.01$, $***p < 0.001$.

### 4.1. Research Question 1: Adverse Patient Outcomes

Each of the three patient outcomes (medication errors, falls, and UTIs) was analyzed with hierarchical logistic regression, using a series of five models as shown in Table 4. Patient dependency was excluded from the analyses due to its lack of bivariate correlation with any of the outcome variables. The non-significant nurse characteristics were also excluded from these analyses due to non-significance in the regression results and to increase the power and parsimony of the models.

**Table 4.** Results of hierarchical logistic regression analyses for three patient outcomes.

| Patient Outcomes and Predictor Variables | Model 1 | Model 2 | Model 3 | Model 4 | Model 5 |
|---|---|---|---|---|---|
| **MEDICATION ERRORS**[1] | OR (95% CI) | OR (95% CI) | OR (95% CI) | OR (95% CI) | OR (95% CI) |
| Patient–RN Ratio | 1.11 * (1.02, 1.21) | 1.14 ** (1.04, 1.25) | 1.13 * (1.03, 1.24) | 1.11 * (1.01, 1.21) | 1.10 * (1.00, 1.21) |
| Patient Acuity [2] | | 5.83 ** (2.04, 16.70) | 4.13 * (1.41, 12.09) | 4.20 ** (1.43, 12.34) | 4.23 ** (1.44, 12.46) |
| Heavy Workload [3] | | | 2.38 ** (1.30, 4.34) | 1.67 (0.88, 3.16) | 1.52 (0.79, 2.93) |
| Interruptions [4] | | | 2.12 * (1.15, 3.89) | 1.75 (0.93, 3.26) | 1.63 (0.86, 3.07) |
| Tasks Left Undone | | | | 1.22*** (1.10, 1.35) | 1.19 ** (1.07, 1.32) |
| Compromised Standards [5] | | | | | 1.66 (0.82, 3.33) |
| Nagelkerke $R^2$ | 2.3% | 8.8% | 16.1% | 21.6% | 22.3% |
| Correct Classification | 86.8% | 86.8% | 86.8% | 86.8% | 87.6% |
| **PATIENT FALLS** [6] | OR (95% CI) | OR (95% CI) | OR (95% CI) | OR (95% CI) | OR (95% CI) |
| Patient–RN Ratio | 1.22 *** (1.11, 1.35) | 1.27 *** (1.15, 1.41) | 1.27 *** (1.14, 1.41) | 1.26 *** (1.12, 1.41) | 1.26 *** (1.12, 1.41) |
| Patient Acuity [2] | | 5.70 ** (1.96, 16.63) | 3.26 * (1.05, 10.12) | 3.32 * (1.06, 10.41) | 3.32 * (1.06, 10.40) |
| Heavy Workload [3] | | | 5.58 *** (2.85, 10.95) | 3.84 *** (1.89, 7.83) | 3.86 *** (1.86, 8.01) |
| Interruptions [4] | | | 2.98 ** (1.46, 6.09) | 2.36 * (1.12, 4.95) | 2.37 * (1.12, 5.01) |
| Tasks Left Undone | | | | 1.32 *** (1.17, 1.49) | 1.32 *** (1.16, 1.50) |
| Compromised Standards [5] | | | | | 0.97 (0.44, 2.16) |
| Nagelkerke $R^2$ | 7.6% | 13.7% | 31.5% | 39.5% | 39.5% |
| Correct Classification | 88.5% | 88.5% | 89.2% | 89.6% | 89.6% |
| **URINARY TRACT INFECTIONS** [7] | OR (95% CI) | OR (95% CI) | OR (95% CI) | OR (95% CI) | OR (95% CI) |
| Patient–RN Ratio | 1.16 ** (1.06, 1.26) | 1.19 *** (1.09, 1.31) | 1.18 ** (1.07, 1.29) | 1.16 ** (1.06, 1.28) | 1.17 ** (1.06, 1.28) |
| Patient Acuity [2] | | 3.95 ** (1.62, 9.64) | 2.50 (0.98, 6.35) | 2.47 (.97, 6.28) | 2.48 (0.98, 6.30) |
| Heavy Workload [3] | | | 3.51 *** (1.92, 6.41) | 2.79 ** (1.49, 5.23) | 2.98 ** (1.56, 5.66) |
| Interruptions [4] | | | 2.53 ** (1.35, 4.72) | 2.21 * (1.17, 4.17) | 2.32 * (1.22, 4.40) |
| Tasks Left Undone | | | | 1.14 ** (1.04, 1.26) | 1.16 ** (1.04, 1.29) |
| Compromised Standards [5] | | | | | 0.72 (0.34, 1.52) |
| Nagelkerke $R^2$ | 4.5% | 9.1% | 21.8% | 24.3% | 24.5% |
| Correct Classification | 86.5% | 86.5% | 87.2% | 88.3% | 88.1% |

Note: [1] 0 = less than weekly, 1 = weekly or more often; [2] 0 = not at all or somewhat acute, 1 = moderately or very acute; [3] 0 = never to a few times a week, 1 = more than a few times a week; [4] 0 = less than every day, 1 = every day or almost every day; [5] 0 = less than every day, 1 = every day; [6] 0 = less than weekly,

1 = weekly or more often; [7] 0 = less than weekly, 1 = weekly or more often. *$p < 0.05$, ** $p < 0.01$, ***$p < 0.001$. Medication Errors $X^2(6) = 58.36$, $p < 0.001$; Patient Falls $X^2(6) = 101.84$, $p < 0.001$; UTIs $X^2(6) = 65.17$, $p < 0.001$.

Higher patient–RN ratios were weakly associated with all three adverse patient outcomes, with odds ratios ranging 1.10 to 1.26 in the final models. Higher patient acuity was associated with medication errors and patient falls, but became non-significant for UTIs in Model 3 after accounting for other variables. Results for Model 3 showed that after accounting for RN staffing levels and patient acuity, perceptions of frequent, heavy perceived nurse workloads and frequent interruptions were strong, independent predictors for all three adverse patient outcomes. For example, nurses who experienced heavy workloads on a daily basis were almost six times more likely to report patient falls on a weekly basis than nurses who experienced heavy workloads less frequently, OR = 5.58, 95% CI (2.85, 10.95). Similarly, nurses who experienced interruptions on a daily basis were three times more likely to report patient falls on a weekly basis than nurses who experienced interruptions less frequently, OR = 2.98, 95% CI (1.46, 6.09). Leaving tasks undone added to the prediction of the three patient outcomes in Model 4 after accounting for RN staffing levels, patient acuity, and nurse perceptions of heavy workload and interruptions to workflow, but with smaller odds ratios ranging from 1.14 (UTIs) to 1.32 (falls). The frequency that professional nursing standards were compromised due to workload did not explain any of the variance in the three patient outcome measures after accounting for other workload factors. In the final model (Model 5), patient acuity was the strongest independent predictor of medication errors, OR = 4.23, 95% CI (1.44, 12.46), whereas perceptions of frequent heavy workloads was the strongest independent predictor of patient falls and urinary tract infections (ORs of 3.86 and 2.98, respectively).

### 4.2. Research Question 2: Nurse Outcomes

Logistic regression results showed that after accounting for individual characteristics and RN staffing levels (non-significant predictors in the final model), patient acuity, perceptions of frequent heavy perceived nurse workloads, frequent interruptions to workflow, leaving tasks undone, and compromised standards were all independent predictors of emotional exhaustion in the final model (Table 5). Model 6 results show that nurses who experienced heavy workloads on a daily basis were three and a half times more likely to report high emotional exhaustion than nurses who experienced heavy workloads less frequently, OR = 3.60, 95% CI (1.94, 6.68). The strongest predictor of emotional exhaustion, after accounting for individual characteristics and the five other workload factors, was compromised professional nursing standards due to workload, with an odds ratio of 4.42, 95% CI (1.86, 10.50).

Multiple regression results showed similar results for nurses' job satisfaction. After accounting for individual characteristics, RN staffing levels and patient acuity, perceptions of heavy perceived nurse workload and frequent interruptions were independently associated with lower levels of job satisfaction in Model 4 (β = −0.28, $p < 0.001$, and β = −0.18, $p < 0.001$), respectively (Table 6). Leaving nursing tasks undone due to workload explained further variation in job satisfaction, as did compromised professional nursing standards. As with emotional exhaustion, compromised professional nursing standards on a daily basis was the strongest predictor of job satisfaction (β = −0.27, $p < 0.001$).Interruptions to workflow ceased to be an independent predictor in Model 6 after compromising standards was added to the equation.

**Table 5.** Results of hierarchical logistic regression analyses for emotional exhaustion [1] among nursing staff.

| Predictor Variables | Model 1 OR (95% CI) | Model 2 OR (95% CI) | Model 3 OR (95% CI) | Model 4 OR (95% CI) | Model 5 OR (95% CI) | Model 6 OR (95% CI) |
|---|---|---|---|---|---|---|
| Age | 1.00 (0.99, 1.02) | 1.00 (0.99, 1.02) | 1.00 (0.99, 1.02) | 1.00 (0.98, 1.01) | 1.00 (0.98, 1.02) | 1.00 (0.98, 1.02) |
| Professional Designation [2] | 0.69 (0.44, 1.08) | 0.75 (0.47, 1.18) | 0.62 (0.39, 1.01) | 0.57 * (0.34, 0.95) | 0.65 (0.38, 1.10) | 0.66 (0.39, 1.14) |
| Employment Status [3] | 0.66 * (0.45, 0.97) | 0.66 * (0.45, 0.97) | 0.700 (0.47, 1.03) | 0.72 (0.47, 1.11) | 0.66 (0.42, 1.03) | 0.68 (0.43, 1.06) |
| Patient–RN ratio | | 1.06 (1.00, 1.12) | 1.07 * (1.01, 1.14) | 1.07 (1.00, 1.14) | 1.04 (0.97, 1.12) | 1.04 (0.97, 1.12) |
| Patient Acuity [4] | | | 2.47 *** (1.54, 3.95) | 1.87 * (1.13, 3.11) | 1.87 * (1.10, 3.17) | 1.82 * (1.07, 3.13) |
| Heavy Workload [5] | | | | 5.62 *** (3.14, 10.06) | 4.07 *** (2.22, 7.46) | 3.60 *** (1.94, 6.68) |
| Interruptions [6] | | | | 2.48 *** (1.56, 3.95) | 1.98 ** (1.22, 3.22) | 1.76 * (1.07, 2.91) |
| Tasks Left Undone | | | | | 1.25 *** (1.15, 1.36) | 1.22 *** (1.11, 1.33) |
| Compromised Standards [7] | | | | | | 4.42 ** (1.86, 10.50) |
| Nagelkerke $R^2$ | 2.2% | 3.2% | 7.4% | 26.0% | 32.9% | 36.0% |
| Correct Classification | 58.2% | 59.3% | 63.5% | 69.0% | 71.2% | 71.5% |

Note: [1] 0 = no burnout, 1 = burnout; [2] 0 = LPN, 1 = RN; [3] 0 = full-time; 1 = part-time or casual; [4] 0 = not at all or somewhat acute, 1 = moderately or very acute; [5] 0 = never to a few times a week, 1 = more than a few times a week; [6] 0 = less than every day, 1 = every day or almost every day; [7] 0 = less than every day, 1 = every day; * $p < 0.05$, ** $p < 0.01$, *** $p < 0.001$. Emotional Exhaustion $X^2(9) = 141.44$, $p < 0.001$.

**Table 6.** Results of hierarchical multiple regression analyses for job satisfaction among nursing staff.

| Predictor Variables | Model 1 β (95% CI) | Model 2 β (95% CI) | Model 3 β (95% CI) | Model 4 β (95% CI) | Model 5 β (95% CI) | Model 6 β (95% CI) |
|---|---|---|---|---|---|---|
| Age | −0.07 (−0.03, 0.01) | −0.07 (−0.03, 0.01) | −0.07 (−0.03, 0.01) | −0.03 (−0.02, 0.01) | −0.05 (−0.03, 0.01) | −0.03 (−0.02, 0.01) |
| Professional Designation [1] | −0.07 (−0.83, 0.13) | −0.10 (−0.97, 0.00) | −0.08 (−0.88, 0.10) | −0.07 (−0.80, 0.12) | −0.09 * (−0.90, −0.01) | −0.10 * (−0.91, −0.06) |
| Employment Status [2] | −0.03 (−0.56, 0.27) | −0.03 (−0.56, 0.26) | −0.04 (−0.60, 0.22) | −0.06 (−0.67, 0.10) | −0.05 (−0.61, 0.14) | −0.06 (−0.63, 0.09) |
| Patient–RN Ratio | | −0.14 ** (−0.16, −0.03) | −0.15 ** (−0.16, −0.04) | −0.12 ** (−0.14, −0.02) | −0.09 * (−0.12, −0.00) | −0.08 (−0.11, 0.00) |
| Patient Acuity [3] | | | −0.11 * (−1.03, −0.06) | −0.03 (−0.64, 0.29) | −0.03 (−0.59, 0.32) | −0.02 (−0.53, 0.34) |
| Heavy Workload [4] | | | | −0.28 *** (−1.88, −0.98) | −0.21 *** (−1.50, −0.60) | −0.16 *** (−1.25, −0.36) |
| Interruptions [5] | | | | −0.18 *** (−1.22, −0.38) | −0.12 ** (−0.97, −0.14) | −0.08 (−0.77, 0.05) |
| Tasks Left Undone | | | | | −0.26 *** (−0.25, −0.12) | −0.19 *** (−0.20, −0.07) |
| Compromised Standards [6] | | | | | | −0.27 *** (−2.05, −1.02) |
| Change in $R^2$ | 0.8% | 1.8% ** | 1.0% * | 13.3% *** | 5.6% *** | 5.6% *** |
| $R^2$ | 0.8% | 2.5% | 3.6% | 16.9% | 22.4% | 28.0% |

Note: [1] 0 = LPN, 1 = RN; [2] 0 = full-time, 1 = part-time or casual; [3] 0 = not at all or somewhat acute, 1 = moderately or very acute; [4] 0 = never to a few times a week, 1 = more than a few times a week; [5] 0 = less than every day, 1 = every day or almost every day; [6] 0 = less than every day, 1 = every day. * $p < 0.05$, ** $p < 0.01$, *** $p < 0.001$. Model 6: $F (9, 442) = 19.13$, $p < 0.001$.

*4.3. Research Question 3: Adverse Patient Outcomes and Mediation Effects*

Mediator variables explain the pathway by which a predictor variable influences an outcome. The regression results presented in Table 4 show that the coefficients for heavy workload and interruptions to workflow decreased in Models 4 and 5 after adding tasks undone and compromised professional nursing standards. These findings suggest that the latter two variables may mediate the relationship between the two predictors, heavy workload and interruptions to workflow, and the three patient outcomes. We tested these effects by running another four series of regressions per Baron and Kenny's recommendations [34] to determine whether leaving nursing tasks undone and compromised professional nursing standards mediated the effects of heavy perceived nurse workload and frequent interruptions on: (a) medication errors; (b) patient falls; and (c) UTIs. When doing a mediation analysis with a dichotomous mediator, the resulting coefficients need to be comparable in terms of their scale. For this reason, Preacher and Leonardelli's Sobel Test analyses [32] were used after the coefficients were treated as per Herr's recommendations [33].

Our results (Table 7) indicate that leaving nursing tasks undone mediated the effects of heavy perceived nurse workload and frequent interruptions on the three patient outcomes ($p < 0.001$). Although leaving nursing tasks undone had a partial mediating effect on patient falls and UTIs, it fully mediated the relationship between both predictors and medication errors. The full mediation is indicated by the non-significant beta coefficient for interruptions in Model 5 of Table 4. The three non-significant coefficients associated with compromising standards (Table 4) show that this predictor failed to meet the first mediation requirement; subsequently, the mediation effect of this predictor was not examined further.

**Table 7.** Sobel test results for mediation effects.

| Outcome and Mediator Variables | Sobel Test Statistic | SE |
|---|---|---|
| **Medication Errors** | | |
| Tasks Left Undone as a mediator of Heavy Workload | 4.5046 *** | 0.1284 |
| Tasks Left Undone as a mediator of Interruptions | 3.6657 *** | 0.0991 |
| **Patient Falls** | | |
| Tasks Left Undone as a mediator of Heavy Workload | 4.7126 *** | 0.1549 |
| Tasks Left Undone as a mediator of Interruptions | 3.7752 *** | 0.1216 |
| **UTIs** | | |
| Tasks Left Undone as a mediator of Heavy Workload | 4.0605 *** | 0.1163 |
| Tasks Left Undone as a mediator of Interruptions | 3.4142 *** | 0.0869 |
| **Emotional Exhaustion** | | |
| Tasks Left Undone as a mediator of Heavy Workload | 5.0062 *** | 0.1217 |
| Tasks Left Undone as a mediator of Interruptions | 3.9216 *** | 0.0977 |
| Compromised Standards as a mediator of Heavy Workload | 2.6055 ** | 0.0288 |
| Compromised Standards as a mediator of Interruptions | 2.7110 ** | 0.0335 |
| **Job Satisfaction** | | |
| Tasks Left Undone as a mediator of Heavy Workload | −4.3700 *** | 0.0890 |
| Tasks Left Undone as a mediator of Interruptions | −3.5921 *** | 0.0680 |
| Compromised Standards as a mediator of Heavy Workload | −3.0952 ** | 0.0230 |
| Compromised Standards as a mediator of Interruptions | −3.2767 ** | 0.0262 |

Note: SE = Standard Error, ** $p \leq 0.01$, *** $p \leq 0.001$.

*4.4. Research Question 4: Nurse Outcomes and Mediation Effects*

The regression results presented in Tables 5 and 6 indicate that the coefficients for heavy workload and interruptions to workflow decreased in Models 5 and 6 after adding leaving nursing tasks undone and compromised professional standards. These findings suggest that the latter two variables may mediate the relationship between two predictors, heavy workload and interruptions

to workflow, and two nurse outcomes (i.e., job satisfaction and emotional exhaustion). We tested these effects as per the third research question.

Our results (see Table 7) demonstrate that the number of nursing tasks left undone partially mediated the effects of heavy perceived nurse workload and frequent interruptions on emotional exhaustion ($p < 0.001$). Compromised professional nursing standards also functioned as a mediator, partially explaining the effects of perceptions of high nurse workload and frequent interruptions on emotional exhaustion ($p < 0.01$) after accounting for the effects of nursing tasks left undone. Similar results were obtained with respect to job satisfaction, except that compromised professional nursing standards was found to fully mediate the effect of interruptions on job satisfaction after accounting for leaving nursing tasks undone (as indicated by the non-significant beta coefficient for interruptions in Model 6 of Table 6).

## 5. Discussion

This study drew on cross-sectional survey data from 472 acute care nurses from one Canadian province. We considered seven indicators of workload: RN staffing levels, patient acuity and patient dependency, nurses' perceptions of heavy workload, nursing tasks left undone, compromised professional nursing standards, and interruptions to workflow. Similar to other research, patient acuity was found to be strongly associated with each of the three adverse patient outcomes [35] and RN staffing levels showed a weaker association [26,36]. Patient dependency was not found to be associated with patient or nurse outcome measures. This may be because patient dependency in this study reflected activities of daily living only. In reality, patient dependency may reflect expanded aspects of patient functionality. In addition, within many acute care contexts in BC, patient activities of daily living are managed by non-nurses. Patient acuity refers to characteristics such as complexity and unpredictability that require nurse surveillance and intervention [30].

After accounting for unit-level workload measures, patient acuity and RN staffing levels, nurse perceptions of frequent, heavy workloads and interruptions to work flow showed strong associations with two patient outcomes, falls and UTIs, and a more modest association with the frequency of medication errors. This study's heavy workload measure includes items associated with nurse perceptions of time pressure, or not enough time to get work done (e.g., arriving early/leaving late, missing breaks, too much work to do). In one simulated study of nurses' decision-making performance, time pressure negatively influenced nurses' capacity to detect the need for intervention, resulting in failure to rescue [37]. Of note is that under conditions without time pressure, nurses with clinical expertise performed better than novice nurses; the positive effects of clinical expertise, however, were negated when time pressure was introduced to clinical simulations [37]. The European Nurses' Early Exit study surveyed over 61,000 nurses [38]. The survey included intent to leave questions, actual turnover and work-related and personal reasons for leaving. The main work-related reason to leave was "time pressure", chosen as the primary work factor for 70% of the sample population. Our findings suggest that nurses are aware of harmful outcomes associated with time pressure; they may compensate for these job-level heavy workload demands by coming in early, staying late and working through breaks.

At the task-level, interruptions divert nurses from their planned activities [39] resulting in decreased performance [40] and increased patient adverse events, such as medication errors [41]. Whether at the job-level (i.e., heavy workload demands) or at the task-level (i.e., interruptions), deleterious consequences from these workload factors can be averted through administrative actions such as implementation of nurse resource teams to cover shift changes and break times [42]; and work redesign initiatives that designate dedicated time for essential tasks, such as medication preparation [43].

Tasks left undone, either partially or fully, mediated the relationships between two workload factors (i.e., perceptions of heavy workloads, interruptions) and patient outcomes. Ball et al. found that care left undone was strongly associated with nurse perceptions of quality, safe care delivery, suggesting that care left undone is a leading, job-level indicator for unsafe staffing [5]. Although unit-level measures, such as staffing adequacy, add to our appreciation of workload demands, job-level

measures, such as leaving tasks undone, may provide administrators with a more accurate depiction of how nurses gauge effective workload management.

With respect to nurse outcomes, patient acuity was associated with higher emotional exhaustion, but it did not influence job satisfaction. A major source of emotional exhaustion is heavy workload demands that are often outside the control of nurses; nurses have "too little time and too few resources to accomplish the job" [44] (p. 260). For job satisfaction, however, a systematic review of hospital nurse job satisfaction found that nurses derived satisfaction from interesting and rewarding work [45]. Care of high acuity patients, therefore, may satisfy nurses by optimizing their professional competencies. In their human factors study of nurses' workloads, Holden et al. found that at the task-level, there were internal and external types of workload demands [7]. External demands, such as interruptions and divided attention, were associated with nurse reports of increased patient safety concerns. Internal demands, such as mental concentration and problem-solving, were not associated with nurses' concerns for patient safety outcomes. As stated by Holden et al. "Perhaps in nursing, some amount of this [mental effort] makes work more satisfying, buffers against burnout and improves patient outcomes through superior performance" [7] (p. 21). Administrators, therefore, need to differentiate between external and internal workload demands; their focus should be on reductions of external factors, such as interruptions, that have deleterious effects on nurses.

After accounting for RN staffing levels and patient acuity, nurses' perceptions of frequent heavy workloads and interruptions were independent predictors of emotional exhaustion. For job satisfaction, perception of frequent heavy workloads was a significant predictor. Baethge and Rigotti found that work interruptions had negative effects on nurses' satisfaction with their performance and their irritation with work [39]. Work irritation is a concept associated with emotional and cognitive strain [46]. Cross-sectional and longitudinal studies have shown that irritation mediates the relationship between workplace stressors and eventual decreases in well-being [47,48]. Baethge and Rigotti further found that time pressure and mental demands fully or partially mediated the relationships between work interruptions and satisfaction with performance [39].

There is evidence, therefore, that job-level heavy workload demands and task-level interruptions involve externally imposed time pressures and mental exertion that negatively influence patient and nurse outcomes. As stated by Baethge and Rigotti, research on workplace demands and stressors is adding to "promising directions for interventions in the field of occupational health promotion" [39] (p. 59). Administrators need to work in collaboration with occupational health and safety officers to utilize best practices that reduce damaging workload factors. Proactive strategies for work interruptions were mentioned above. Health circles are an intervention to address the mental and emotional strain of workloads [49]. Health circles are workplace discussion groups where employees are encouraged to discuss and identify opportunities to decrease workload demands—giving control to employees who are the experts in their workplace.

A significant finding from our study was that the strongest predictor of both nurse outcomes (i.e., emotional exhaustion and job satisfaction) was compromised professional nursing standards due to workload. Moreover, compromised standards were also found to be a significant mediator of both heavy perceived workload and interruptions for both nurse outcomes. Mediation testing is used to test hypothesized casual chains where predictor variables influence intervening variables (i.e., the mediator) that, in turn, influence outcome variables. If the predictor variable influences the outcome variable only through the mediator variable (i.e., indirectly), this is considered full mediation. On the other hand, if the predictor variable influences the outcome variable directly and indirectly through the mediator variable there is partial mediation. In this instance, our findings suggest that heavy workloads and interruptions influence nurse outcomes both directly and indirectly through the mediator variables (i.e., nursing tasks left undone, and compromised standards).

Nursing is a caring profession built upon nurse-patient relationships. When nursing is reduced to "task and time" mechanistic approaches to care delivery, nurses suffer from emotional and moral distress [50,51]. Compromised nursing standards are a source of emotional distress and moral distress, with deeper ethical roots. "…moral distress occurs when the internal environment of nurses—their values and perceived obligations—are incompatible with the needs and prevailing

views of the external work environment" [52] (p. 1). Outcomes from emotional and moral distress include emotional exhaustion/burnout, job dissatisfaction and eventual exit from the profession [52–55]. Epstein and Delgado [52] recommended that administrators engage nurses in discussions around values conflicts, while Pendry [56] advocated for informal team discussions and formal ethics committees.

Van Bogaert et al. studied the relationships between the nurse practice environment and job outcomes and nurse-assessed quality of care [31]. Job outcomes included job satisfaction, intent to stay in the hospital, and intent to stay in nursing. Mediators included nurse perceptions of workload; decision latitude (i.e., ability to make decisions and use personal/professional skills); social capital (i.e., shared values and perceived team/organizational trust); and three dimensions of burnout (i.e., emotional exhaustion, depersonalization and personal accomplishment). There were direct and indirect effects for workload on job outcomes. Workload, decision latitude and social capital mediated the relationship between practice environment and outcomes variables via the burnout variables. A key finding was that unit-level nursing management had a strong, direct impact on the study's outcomes. The researchers concluded that unit-level leaders, in particular, can influence job outcomes and nurse perceptions of quality of care by monitoring and responding to nurses' workload demands, involving nurses in decisions related to patient care delivery, and promoting shared professional values among interdisciplinary team members.

In our study, we examined two potential mediators with respect to patient outcomes, and four potential mediators with respect to nurse outcomes. Tasks left undone was found to be a significant mediator of perceived heavy workload and interruptions for all three patient outcomes, suggesting that in addition to their direct effects, heavy workload and interruptions influence patient outcomes indirectly through their influence on nurses' ability to complete essential tasks. Similarly, we found indirect effects from perceived heavy workloads and interruptions on both nurse outcomes through tasks left undone and compromised professional nursing standards. These two mediators, therefore, should serve as critical indicators for administrators to monitor and track: these mediators may be the "litmus test" for nurses' capacity to effectively deliver care within their work environments. Nursing is a unique profession where essential tasks left undone and compromised professional standards signify the potential for adverse patient and nurse outcomes.

*Study Limitations*

A major strength of this study was that its sample consisted of both RNs and LPNs drawn from multiple hospitals across the four largest health authorities in the province. In Canada and globally, a trend in health care is to use teams of RNs and LPNs to deliver patient care. Health services research, therefore, needs to include the perspectives of RNs and LPNs [57]. Second, the explanatory model included seven indicators of workload so that independent effects of each could be investigated. However, causal inferences are limited by the cross-sectional data. Other limitations are the low response rate and inconsistency in the time dimension of the some of the measures used in the study. For example, nursing tasks left undone were measured over the last shift, but patient adverse events were measured over the last year and later recoded as less than weekly versus weekly or more often. This inconsistency may have confounded the study findings. Asking nurses' perceptions of a phenomenon over the last year or last month also increases the possibility of measurement error due to recall bias. The low response rate of the study leads to concerns of sample bias and generalizability of the findings. High response rates, however, do not guarantee representation and vice versa: researchers need to look beyond survey response rates to factors such as non-response error. Non-response error occurs when a significant number of people in the survey sample do not respond and have different characteristics from those who do respond [58]. As cited in Havaei et al., the total study sample was compared with Canadian Institute for Health Information reports of provincial nurse demographics [59]. We found that this study sample is similar to the BC nursing workforce with respect to age, gender, and employment status [60].

## 6. Conclusions

As we explore those aspects of nurses' work environments associated with workload demands, we need to recognize how different levels of workload demands have differential effects on patient and nurse outcomes. Overall, this study demonstrated that job-level nurse perceptions of heavy workloads and task-level interruptions adversely influence patient and nurse outcomes. Other research suggests that externally imposed time pressures and mental demands may be part of causal pathways that require further explication. We discovered that two important mediators are nurse reports of tasks left undone and compromised professional nursing standards. Although tasks left undone was a mediator for both patient and nurse outcomes, compromised professional nursing standards only mediated nurse outcomes, denoting perhaps, how compromised nurse values matter significantly to this caring profession.

Nurses' workloads are often evaluated with respect to unit-level staffing adequacy and/or patient acuity systems. Quantitative measures, such as these, contribute to our appreciation of nurse workload demands, but they exclude many complex, invisible aspects of nurses' work that can only be gauged by nurses themselves. Proactive reduction of patient adverse events, nurse emotional exhaustion and decreased job satisfaction, therefore, requires healthcare administrators to collaboratively address those factors in the work environment that impact nurse workloads.

**Author Contributions:** All three authors contributed to the conceptual design of the study, data analyses and interpretation of findings. Dr. MacPhee was the principal investigator on the study, accountable for ethics approval, study planning and implementation. Dr. MacPhee's area of expertise is nurse work environments and safe staffing: she was the lead author on the Background and Discussion. Dr. Dahinten conceptualized and conducted the statistical analysis, and was the lead author for the Methods and Findings sections of the paper. Dr. Havaei assisted with analysis and writing.

**Acknowledgement:** The authors would like to acknowledge the funding received from British Columbia Nurses Union and the Collaborative Alliances for Nursing Outcomes to support the study.

**Conflicts of Interest:** The authors declare no conflict of interest.

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
