# Peer review of "The Impact of Heavy Perceived Nurse Workloads on Patient and Nurse Outcomes"

_admsci, doi:10.3390/admsci7010007_

Round 1

Reviewer 1 Report

Overall this manuscript is well written to deliver the study conducted by the authors, which was the identification of relationships among nursing workload-related factors, patient outcomes, and nurse outcomes.

Let me suggest the authors some recommendations to clarify and improve the manuscript:

Line 31 – The authors started the sentence, “The global RN4CAST project with over 11,000 patients…” Before this sentence, it would be good to introduce the RN4CAST a little bit. For example, it would be better to add a sentence, ‘The global RN4CAST consortium (http://www.rn4cast.eu/en/consortium.php) has been established and coordinated to support the accuracy of forecasting models and generate new approaches to more effective management of nursing resources across countries.’

Line 39-41 – As the authors didn’t use the terms - ‘leading indicators’ and ‘lagging indicators'-in the later of the manuscript at all, this sentence would not be necessary.

Line 58 – The sentence, “Assessment of patient care needs underpins this approach to workload’ needs to be clearer. It could be stated as ‘Assessment of patient care needs underpins the nursing workload measurement.’

Line 60-62 – I would edit the sentence, “Nurse assessment can be a reliable, valid approach for making determinations of patient acuity and dependency needs” as ‘Nursing assessment has been used to determine patient acuity and dependency needs.’

Line 76 – The sentence,” Nurses reported a mean of four missed care items,” would need to provide what four missed care items for the reader.

Line 100-101 – It is stated that “….burnout was positively associated with unit-level staffing adequacy and task-level external demands, such as interruptions [7].” The nurse burnout might be positively associated with unit-level staffing inadequacy rather than its adequacy. It is recommendable for the authors to double-check it with the cited article.

Line 104-106 – The sentence, “Nurse reports of patient adverse events…, because accurate unit-level data are difficult to obtain.” needs to clearer. Was it about nurses’ recall of patient adverse events? Why could nurse reports of patient adverse events be just as a proxy for the unit-level data rather than be the actual data? It would be better to clarify if the data were nurses’ recall or nurses’ perceptions, or nurses’ reports. This clarification is very important for the reader to evaluate the reliability of such data.

Line 106-107 – The sentence could be revised to clarify the fact by adding or revising such as ‘Some research has established [a] concordance between nurse reports estimates and [actual] patient adverse events, such as falls with injuries [27].’

Line 114 - In order to clarify each of the seven workload factors, the phrase of patient acuity and dependency needs to be separated as patient acuity and patient dependency. Please do the same through the manuscript.

Line 166-167 – Regarding RN Staffing Levels, the authors explained that “Patient-to-RN ratio was used rather than the patient-to-nurse (RN or LPN) ratio for consistency purposes, as many units did not utilize LPNs.” However, 25% of the sample was the LPN in the study (LPNs=118, RNs=354). So, the use of ‘Patient-to RN ratio’ should be indicated as one of the study limitations later in the manuscript.

Line 183-185 – Regarding Nursing Tasks Left Undone, the authors asked nurses to recall them during their most recent shift. Other measures asked nurses to recall or consider the measured during the last year, but which turned out to be weekly or daily data per nurses’ recall or perception. The inconsistency in the units of the measures might have worked as a confounding factor. The authors need to indicate a potential impact due to the inconsistency in the units across the measures with nurses’ perception based on their recall on the statistical analyses could be considered as one of the study limitations later in the manuscript.

Line 213 – The authors stated that “Among the RNs, 75% had a nursing degree.” What does this mean? It is hard to find a relevant data in Tables.

Line 330-333 – It is recommended to revise the orginal sentence, “This study drew on cross-sectional data from 472 acute care nurses from one Canadian province” to a sentence, ‘This study drew on a cross-sectional survey data with 472 acute care nurses from one Canadian province.’

Line 335-339 – The authors stated that “Patient dependency was not found to be associated with patient or nurse outcome measures. This may be due to the fact that dependency is related to activities of daily living, and in many care environments, these activities are performed by non-nurses.” It is not a solid rationale for the non-significant relationship between patient dependency and patient or nurse outcomes. So, it is recommended to revise it. An example sentence could be ‘Patient dependency was not found to be associated with patient or nurse outcomes. This result may be due to the fact that patient dependency in this study considered the aspect of activities of daily living only. Patient dependency could be measured with expanded aspects of patient functionality, and its relationship with patient or nurse outcomes could then be different.’

Line 345-347 – The sentence, “In one simulated study of nurses’ decision-making performance, time pressure negatively influenced nurses’ capacity to detect the need for intervention, resulting in failure to rescue” needs a citation.

It is recommended to reorganize or revise the entire section of Discussion, considering the mediation relationships among the variables. It is believed that the discussion about the mediating relationships is so critical to understand the factors related to nursing workload and to devise appropriate strategies for nursing workforce management.

Also, a question is raised if the authors have tried to enter the variables in a different order in the hierarchical logistic regression analyses. Upon the entering order of the variables, this type of analysis could provide interesting results. After iterative analyses, the authors could then get more confirmative results about the mediation relationships of the variables. In addition, the use of structural equation modeling is recommended to extract direct and indirect relationships among the variables. It would provide a clearer picture of the mediation relationships, not requiring repeated hierarchical regression analyses in different entering orders with the variables.  

It is strongly recommended to double-check the use of tense throughout the manuscript.

It is a great study. Hope the authors would continue relevant studies to improve nursing workforce management.

Author Response

Thank you so much for your careful review of our manuscript. Please see below for our response to your review (also attached).

Reviewer 1

Line 31 – The authors started the sentence,   “The global RN4CAST project with over 11,000 patients…” Before this sentence,   it would be good to introduce the RN4CAST a little bit. For example, it would   be better to add a sentence, ‘The global RN4CAST consortium   (http://www.rn4cast.eu/en/consortium.php) has been established and   coordinated to support the accuracy of forecasting models and generate new   approaches to more effective management of nursing resources across   countries.’

Your suggestion has been added to the   manuscript.  Thank you.

Line 39-41 – As the authors didn’t use the   terms - ‘leading indicators’ and ‘lagging indicators'-in the later of the   manuscript at all, this sentence would not be necessary.

We used the terms, “predictors” and   “outcomes” in the rest of our manuscript, so we changed the terms in Lines   39-41 to be consistent. We kept the following sentence on leading and lagging   indicators from Ball et al. (2014), because this is important, related   research.

Our revision:

“We were interested in identifying key predictors   that can be used to identify worrisome trends and avert serious outcomes,   such as patient mortality and morbidity.”

Line 58 – The sentence, “Assessment of   patient care needs underpins this approach to workload’ needs to be clearer.   It could be stated as ‘Assessment of patient care needs underpins the nursing   workload measurement.’

This sentence has been revised as per your   recommendation. 

Line 60-62 – I would edit the sentence,   “Nurse assessment can be a reliable,   valid approach for making determinations of patient acuity and dependency   needs” as ‘Nursing assessment has been used to determine patient acuity and   dependency needs.’

This sentence has been revised as per your   recommendation

Line 76 – The sentence,” Nurses reported a   mean of four missed care items,” would   need to provide what four missed care items for the reader.

Thank you for bringing this to our   attention. We revised the sentence to “On average,   nurses reported leaving four care items undone on their most recent shift.”

Line 100-101 – It is   stated that “….burnout was positively   associated with unit-level staffing adequacy and task-level external demands,   such as interruptions [7].” The nurse burnout might be positively associated   with unit-level staffing inadequacy rather than its adequacy. It is   recommendable for the authors to double-check it with the cited article.

Holden et al (2011) used the term “staffing   adequacy”. Therefore, we also decided to use this term. We clarified the   sentence as:

“Holden et al. found that nurse job satisfaction was   positively associated with a unit-level workload measure, staffing adequacy;   burnout was negatively associated with unit-level staffing adequacy, and   positively associated with task-level external demands, such as interruptions   [7].”   

Line 104-106 – The sentence, “Nurse reports   of patient adverse events…, because accurate unit-level data are difficult to   obtain.” needs to clearer. Was it about nurses’ recall of patient adverse   events? Why could nurse reports of patient adverse events be just as a proxy   for the unit-level data rather than be the actual data? It would be better to   clarify if the data were nurses’ recall or nurses’ perceptions, or nurses’   reports. This clarification is very important for the reader to evaluate the   reliability of such data.

This sentence has been revised to “Although   nurse reports of patient adverse events are prone to recall bias, some   research has established concordance between nurse reports and actual patient   adverse events, such as falls with injuries [27]. For the RN4CAST studies,   nurse ratings of unit-level quality of care included estimates of frequency   of patient adverse events, such as medication errors, falls and   hospital-acquired infections [6].”

Line 106-107 – The sentence could be   revised to clarify the fact by adding or revising such as ‘Some research has   established [a] concordance between nurse reports estimates and   [actual] patient adverse events, such as falls with injuries [27].’

We think adding a sentence that identified   the possibility of recall bias in nurse reports of patient adverse events is   sufficient here. Also, nurse reports of [patient adverse events] is language   commonly used in the literature. For this reason, we think nurse reports is   more appropriate than nurse estimates. Having said that, we added the term   “actual” to further clarify the difference between nurse reports and actual   patient adverse events. As stated above, this sentence has been revised to “Although nurse reports of patient adverse events are   prone to recall bias,  some research   has established concordance between nurse reports and actual patient adverse   events, such as falls with injuries”

Line 114 - In order to clarify each of the   seven workload factors, the phrase of patient acuity and dependency needs to   be separated as patient acuity and patient dependency. Please do the same   through the manuscript.

Patient acuity and dependency has been   changed to patient acuity and patient dependency throughout the manuscript,   as advised.

Line 166-167 – Regarding RN Staffing   Levels, the authors explained that “Patient-to-RN ratio was used rather than   the patient-to-nurse (RN or LPN) ratio for consistency purposes, as many   units did not utilize LPNs.” However, 25% of the sample was the LPN in the   study (LPNs=118, RNs=354). So, the use of ‘Patient-to RN ratio’ should be   indicated as one of the study limitations later in the manuscript.

While we appreciate the reviewer’s comment,   we do not consider this a limitation of this study. All the study units had   RNs, and a patient-to-RN ratio gave us a consistent measure of staffing   levels across all units. A patient-to-nurse measure is measuring a different   construct, and we did not use this measure because it was not representative   of all study units.

Line 183-185 – Regarding Nursing Tasks Left   Undone, the authors asked nurses to recall them during their most recent   shift. Other measures asked nurses to recall or consider the measured during   the last year, but which turned out to be weekly or daily data per   nurses’ recall or perception. The inconsistency in the units of the measures   might have worked as a confounding factor. The authors need to indicate a   potential impact due to the inconsistency in the units across the measures   with nurses’ perception based on their recall on the statistical analyses   could be considered as one of the study limitations later in the manuscript.

We agree and thank you for bringing this to   our attention. The following sentence was added to ‘Study Limitations”   section:

“Other   limitations are the low response rate and inconsistency in the units of   measures of the some of the measures used in the study. For example, nursing   tasks left undone were measured over the last shift, but patient adverse   events were measured over the last year and later recoded as less than weekly   vs weekly or more often. This inconsistency in the units of the measures may   have confounded the study findings”. 

Line 213 – The authors stated that “Among   the RNs, 75% had a nursing degree.” What does this mean? It is hard to find a   relevant data in Tables.

This sentence was revised as following

“Among the sample, 56% had a nursing degree”.

Line 330-333 – It is recommended to revise   the orginal sentence, “This study drew on cross-sectional data from 472 acute   care nurses from one Canadian province” to a sentence, ‘This study drew on a cross-sectional survey data with 472 acute care nurses from one Canadian province.’

This sentence was revised as following

“This   study drew on cross-sectional survey data from 472 acute care nurses from British   Columbia, Canada.”

Line 335-339 – The authors stated that   “Patient dependency was not found to be associated with patient or nurse   outcome measures. This may be due to the fact that dependency is related to   activities of daily living, and in many care environments, these activities   are performed by non-nurses.” It is not a solid rationale for the   non-significant relationship between patient dependency and patient or nurse   outcomes. So, it is recommended to revise it. An example sentence could be   ‘Patient dependency was not found to be associated with patient or nurse   outcomes. This result may be due to the fact that patient dependency in this   study considered the aspect of activities of daily living only. Patient dependency   could be measured with expanded aspects of patient functionality, and its   relationship with patient or nurse outcomes could then be different.’

This sentence was revised after   consideration of your comments:

“This   may be due to the fact that patient dependency in this study reflected   activities of daily living only. In reality, patient dependency may reflect   expanded aspects of patient functionality. In addition, within many acute   care contexts in BC, patient activities of daily living are managed by   non-nurses.”

Line 345-347 – The sentence, “In one   simulated study of nurses’ decision-making performance, time pressure   negatively influenced nurses’ capacity to detect the need for intervention,   resulting in failure to rescue” needs a citation.

Citation [37] was added at the end of this   sentence.

It is recommended to reorganize or revise   the entire section of Discussion, considering the mediation relationships   among the variables. It is believed that the discussion about the mediating   relationships is so critical to understand the factors related to nursing   workload and to devise appropriate strategies for nursing workforce   management.

We discussed this comment respectfully, and   have chosen to not reorganize the Discussion. However, we added content to   the Discussion in two places to emphasize the importance of the mediating   effects.

See Lines 428-435:  “Mediation testing is   used to test hypothesized casual chains where predictor variables influence   intervening variables (i.e., the mediator) that, in turn, influence outcome   variables. If the predictor variable influences the outcome variable   only through the mediator variable (i.e., indirectly), this   is considered full mediation. On the other   hand, if the predictor variable influences the outcome variable   directly and indirectly through the mediator   variable there is partial mediation. In this   instance, our findings suggest that heavy workloads and interruptions   influence nurse outcomes both directly and indirectly through the mediator variables   (i.e., nursing tasks   left undone, and compromised standards).”

Also,   see lines 459-466: “In our study, we examined two potential mediators with   respect to patient outcomes, and four potential mediators with respect to   nurse outcomes. Tasks left undone was found to be a significant mediator of   perceived heavy workload and interruptions for all three patient outcomes,   suggesting that in addition to their direct effects, heavy workload and   interruptions influence patient outcomes indirectly through their influence   on nurses’ ability to complete essential tasks. Similarly, we found indirect   effects from perceived heavy workloads and interruptions on both nurse   outcomes through tasks left undone and compromised professional nursing   standards.”

Also, a question is raised if the authors   have tried to enter the variables in a different order in the hierarchical   logistic regression analyses. Upon the entering order of the variables, this   type of analysis could provide interesting results. After iterative analyses,   the authors could then get more confirmative results about the mediation   relationships of the variables. In addition, the use of structural equation   modeling is recommended to extract direct and indirect relationships among   the variables. It would provide a clearer picture of the mediation   relationships, not requiring repeated hierarchical regression analyses in   different entering orders with the variables.  

The order of variable entry was chosen   based on our initial hypotheses about possible relationships. We agree that   Structural Equation Modelling (SEM) can provide more confirmatory results   with respect to examining mediation pathways, but since this was an   exploratory study with cross-sectional data, we believe that multiple   regression was an appropriate method of analysis. In addition, our sample   size of 472 was relatively small for SEM with the large number of predictor   variables, and direct and indirect pathways in the model. We agree that   future research should test direct and indirect relationships using SEM,   ideally with a longitudinal research design.

It is strongly recommended to double-check   the use of tense throughout the manuscript.

We have checked tenses throughout the   manuscript.

Reviewer 2 Report

This is an excellent piece of research, and the text used to present it is well crafted and compelling. My only concern with the paper is with respect to the target audience. The depiction of the results in the Table 3 and especially Table 4 requires a high degree of detailed expertise in current statistical practices. For example, Table 4, with its depiction of the 5 different models, would benefit from significantly more discussion and description about  what is being described, if this is to be interpreted by health care practitioners and decision makers. The same would be true of the use of the statistical term mediation, a term not likely familiar to a significant number of practitioners and decision makers. If the target audience is researchers and statisticians, the reporting of the results is likely quite appropriate. However, if this publication is directed at practitioners and decision makers, it would benefit from more description to support the reporting of the statistical processes in the tables. Because of the high degree of value that this study would add to the literature, it would be advisable from my perspective to make it as widely accessible in style as is possible. There are important findings here, and it would be advantageous to share them widely, because of their contribution to the literature and to  health care quality. However, the complexity of the reported results would benefit from more supporting description to increase the understandability, in order to make these results accessible to many people who would benefit form an understanding of the work.

Author Response

Thank you for your positive feedback and your thoughtful review of our work. Please see below for our response to your review (also attached).

Reviewer 2

This is an excellent piece of research, and   the text used to present it is well crafted and compelling. My only concern   with the paper is with respect to the target audience. The depiction of the results in the   Table 3 and especially Table 4 requires a high degree of detailed expertise   in current statistical practices. For example, Table 4, with its depiction of   the 5 different models, would benefit from significantly more discussion and   description about what is being described, if this is to be interpreted   by health care practitioners and decision makers. The same would be true of   the use of the statistical term mediation, a term not likely familiar to a   significant number of practitioners and decision makers. If the target   audience is researchers and statisticians, the reporting of the results is   likely quite appropriate. However, if this publication is directed at   practitioners and decision makers, it would benefit from more description to   support the reporting of the statistical processes in the   tables. Because of the high degree of value that this study would add to   the literature, it would be advisable from my perspective to make it as   widely accessible in style as is possible. There are important findings here,   and it would be advantageous to share them widely, because of their   contribution to the literature and to  health care quality. However, the   complexity of the reported results would benefit from more supporting   description to increase the understandability, in order to make these results accessible   to many people who would benefit form an understanding of the work.

Thank   you for your positive feedback and your thoughtful review of our work.

Our   target audience includes researchers and decision-makers/administrators.   Because of the breadth of our target audience, we provided enough details in   our Methods and Findings section to guide researchers through the study   design, data analyses and key findings.

The   Background and Discussion are intended to offer insights of significance to decision-makers   and administrators. We added further explication of mediation effects as   background for readers who are less familiar with mediation. Please see Lines   427-433 and Lines 457-464.

Reviewer 3 Report

Thanks for the opportunity to review this well-written article.  The fit between research and methods was appropriate.  While the findings were not especially surprising they add to the critical mass of evidence suggesting that there is more complexity to workload than merely nurse to patient ratios. The relationship of the measures of workload used here and the findings reported to the broader literature in management and organizational behavior is not provided--so the audience for these findings in this particular journal may be limited.  The testing of quality of work measures (such as tasks left undone) as mediators of the relationships between workload and the various dependent measures is an interesting and innovative feature of this report.  

Title and throughout:  Would prefer to see "heavy nurse workloads" referred to as "heavy perceived nurse workloads"--because no objective measure is offered here

Line 15: Abstract: Spell out BC and  indicate more clearly that this was a  Canadian sample

Line 102: would be stronger if reference were studies that nursing workloads were associated with adverse patient outcomes like reference 26. Reference 25- is handbook- not studies that support this. I suspect there are primary references in the handbook that should be used.

Line 116: sent has “mediating effects” twice. Needs rewording.

Line 183: Would like a listing of the specific tasks left undone or at least examples

The limitations section is rather short and glosses over the limitations of self-report data, especially, but not restricted to reports of patient outcomes (although some reference is made to the apparent concordance/correlation of nurse reports with objectively measured outcomes early on, not enough detail is presented in limitations) as well as arguable validity of many of the independent measures and common methods bias.  Low response rate and use of a union database (are all nurses unionized in this jurisdiction?) both likely introduce bias into the sample.

Author Response

Thank you for your thoughtful review of our manuscript. Please see below for our response to your review (also attached).

Reviewer 3

Title and throughout:  Would prefer   to see "heavy nurse workloads" referred to as "heavy perceived   nurse workloads"--because no objective measure is offered here

This   change was incorporated in the title and throughout the manuscript.

Line 15: Abstract: Spell out BC and    indicate more clearly that this was a  Canadian sample

This   sentence was revised as following

“This   was a cross-sectional correlational study of 472 acute care nurses from   British Columbia, Canada.”

Line 102: would be stronger if reference   were studies that nursing workloads were associated with adverse patient   outcomes like reference 26.

We   are somewhat confused by this comment as the paragraph containing Line 102   discusses nurse outcomes such as job satisfaction and burnout.

Reference 25- is handbook- not studies   that support this. I suspect there are primary references in the handbook   that should be used.

This   handbook provides an overview/synthesis of many workload studies and their   results, and that is why it is used to support our introductory sentence. We   provide primary references in subsequent text.

Line 116: sent has “mediating effects” twice. Needs rewording.

Thank   you for noting this error. This was fixed as the following

“In   addition, we tested the potential mediating effects of two variables; nursing   tasks left undone and compromised professional nursing standards”.

Line 183: Would like a   listing of the specific tasks left undone or at least examples

Examples   of these nursing tasks were added.

“Thirteen   nursing tasks were identified by Ball et al. including administering   medications on time, preparing patients and families for discharge, and   adequate patient surveillance [5]. We added an “other” option to our survey   tool.

The   limitations section is rather short and glosses over the limitations of   self-report data, especially, but not restricted to reports of patient   outcomes (although some reference is made to the apparent   concordance/correlation of nurse reports with objectively measured outcomes   early on, not enough detail is presented in limitations) as well as arguable   validity of many of the independent measures and common methods bias.    Low response rate and use of a union database (are all nurses unionized   in this jurisdiction?) both likely introduce bias into the sample.

All   direct care nurses in acute care settings in British Columbia are unionized.   A comment was added to the Materials and Methods Section, Lines 155-156, to   reflect this suggestion. See below.  

“All direct care nurses in acute care settings in   British Columbia (BC) are unionized; therefore we had a complete sample   frame.”

The   limitation section was expanded to reflect other recommendations. See below   for revisions on Lines 474-488:

“Other limitations are the low   response rate and inconsistency in the time dimension of the some of the   measures used in the study. For example, nursing tasks left undone were   measured over the last shift, but patient adverse events were measured over   the last year and later recoded as less than weekly versus weekly or more   often. This inconsistency may have confounded the study findings. Asking   nurses’ perceptions of a phenomenon over the last year or last month also   increases the possibility of measurement error due to recall bias. The low   response rate of the study leads to concerns of sample bias and   generalizability of the findings. High response rates, however, do not   guarantee representation and vice versa: researchers need to look beyond   survey response rates to factors such as non-response error. Non-response   error occurs when a significant number of people in the survey sample do not   respond and have different characteristics from those who do respond [57]. As   cited in Havaei et al., the total study sample was compared with Canadian   Institute for Health Information reports of provincial nurse demographics   [58]. We found that this study sample is similar to the BC nursing workforce   with respect to age, gender, and employment status [59].”